# The Longitudinal Changes in Subcutaneous Abdominal Tissue and Visceral Adipose Tissue Volumetries Are Associated with Iron Status

**DOI:** 10.3390/ijms24054750

**Published:** 2023-03-01

**Authors:** Alejandro Hinojosa-Moscoso, Anna Motger-Albertí, Elena De la Calle-Vargas, Marian Martí-Navas, Carles Biarnés, María Arnoriaga-Rodríguez, Gerard Blasco, Josep Puig, Diego Luque-Córdoba, Feliciano Priego-Capote, José María Moreno-Navarrete, José Manuel Fernández-Real

**Affiliations:** 1Nutrition, Eumetabolism and Health Group, Girona Biomedical Research Institute (IdibGi), 17007 Girona, Spain; 2Department of Medical Sciences, School of Medicine, University of Girona, 17003 Girona, Spain; 3Centro de Investigación Biomédica en Red Fisiopatología de la Obesidad y Nutrición (CIEROBN), 28029 Madrid, Spain; 4Medical Imaging, Girona Biomedical Research Institute (IdibGi), 17007 Girona, Spain; 5Department of Radiology (IDI), Dr. Josep Trueta University Hospital, 17007 Girona, Spain; 6Department of Analytical Chemistry, University of Córdoba, Annex Marie Curie Building, Campus of Rabanales, 14014 Córdoba, Spain; 7Consortium for Biomedical Research in Frailty & Healthy Ageing (CIBERFES), Carlos III Institute of Health, 28029 Madrid, Spain

**Keywords:** adipose tissue, obesity, iron metabolism, insulin resistance, inflammation

## Abstract

Excess iron is known to trigger adipose tissue dysfunction and insulin resistance. Circulating markers of iron status have been associated with obesity and adipose tissue in cross-sectional studies. We aimed to evaluate whether iron status is linked to changes in abdominal adipose tissue longitudinally. Subcutaneous abdominal tissue (SAT) and visceral adipose tissue (VAT) and its quotient (pSAT) were assessed using magnetic resonance imaging (MRI), at baseline and after one year of follow-up, in 131 (79 in follow-up) apparently healthy subjects, with and without obesity. Insulin sensitivity (euglycemic– hyperinsulinemic clamp) and markers of iron status were also evaluated. Baseline serum hepcidin (*p* = 0.005 and *p* = 0.002) and ferritin (*p* = 0.02 and *p* = 0.01)) were associated with an increase in VAT and SAT over one year in all subjects, while serum transferrin (*p* = 0.01 and *p* = 0.03) and total iron-binding capacity (*p* = 0.02 and *p* = 0.04) were negatively associated. These associations were mainly observed in women and in subjects without obesity, and were independent of insulin sensitivity. After controlling for age and sex, serum hepcidin was significantly associated with changes in subcutaneous abdominal tissue index (iSAT) (β = 0.406, *p* = 0.007) and visceral adipose tissue index (iVAT) (β = 0.306, *p* = 0.04), while changes in insulin sensitivity (β = 0.287, *p* = 0.03) and fasting triglycerides (β = −0.285, *p* = 0.03) were associated with changes in pSAT. These data indicated that serum hepcidin are associated with longitudinal changes in SAT and VAT, independently of insulin sensitivity. This would be the first prospective study evaluating the redistribution of fat according to iron status and chronic inflammation.

## 1. Introduction

Iron is increasingly recognized as modulating the phenotype of metabolic diseases. Different parameters evaluating iron stores have been associated with obesity status. In an initial study, circulating ferritin was positively associated with visceral and subcutaneous fat areas and negatively associated with the hepatic fat content [1]. Serum ferritin concentration was later confirmed to be associated with several obesity indicators [2,3]. In addition to serum ferritin, serum hepcidin and hepatic iron content (HIC) were increased in subjects with obesity [4]. Out of the 25 studies included in a systematic review of the literature from 2012, only 10 performed a comparative analysis between subjects with and without obesity. Of these ten studies, seven showed that obesity groups had increased mean-hemoglobin concentrations; six had raised serum ferritin; and four had reduced transferrin saturation [5]. All these results are indicative of alterations in iron storage and chronic inflammation in obesity. Interestingly, increased serum ferritin and iron were also associated with decreased markers of adipocyte differentiation [6], and the reduction in iron by phlebotomy led to improved glucose tolerance [7]. Increased adipose-tissue markers of iron accumulation using magnetic resonance imaging were also associated with obesity, insulin resistance and markers of adipose tissue dysfunction [8]. Despite all this information, we found no longitudinal studies testing whether iron status is associated with spontaneous changes in fatness.

There is experimental evidence of the close links between adipose tissue function and iron metabolism. Transferrin, the main circulating protein that binds iron with high affinity, is highly expressed in adipocytes in association with systemic insulin sensitivity, being required for adipogenesis and the maintenance of adipocyte function [8]. In fact, iron chelation blunted adipocyte differentiation in the absence of iron overload, which was recovered after the administration of transferrin [9] or other iron donors such as lactoferrin [10]. Increased body-iron stores have also been found to impact the development of adiposity in animal studies, impairing both adipogenesis and insulin action [11]. Mice with diabetes and obesity exhibited raised accumulation of iron in adipose tissue [12], a situation also associated with impaired adipocyte differentiation [9]. Iron chelation by deferoxamine in this context led to improved adipogenesis and reduced oxidative stress, and improved inflammation and adipocyte hypertrophy, in parallel with improved insulin action [7,12]. Similar findings in both extremes of iron stores suggest that a precise and fine-tuned iron availability is required for adipogenesis. Strengthening the importance of adipose-tissue iron on whole-body metabolism, a recent study demonstrated that low adipocyte-iron levels can restrain intestinal lipid absorption and prevent caloric influx in high fat-fed mice [13].

The evaluation of certain parameters as serum ferritin needs to take into account the inflammatory counterpart of this protein. In fact, serum uric acid, a marker of oxidative stress, is positively associated with VAT and SAT areas [14], while C-reactive protein (CRP) in plasma is also well known for increasing with measurements of abdominal adiposity, including visceral fat [2], and has also been associated with weight gain in some longitudinal studies [15,16].

The findings linking iron metabolism and adiposity are limited by their cross-sectional approach in the majority of studies. Analyzing the associations between the iron-metabolism parameters and the temporal evolution of adipose tissue could establish those indicators as predictive for adipose tissue changes. Therefore, we aimed to evaluate whether different parameters affected by iron exposure were associated with changes in adiposity and insulin action in a longitudinal study, controlling for inflammatory- and oxidative-stress markers. Specifically, we evaluated the temporal variations in subcutaneous and visceral adipose-tissue volumetry and distribution using MRI over one year in subjects with and without obesity. Considering that adrenal- or gonadal-steroid hormones are important metabolism modulators that contribute to body fat distribution [17,18,19] and iron homeostasis [20,21], levels of cortisol, estradiol, dehydroepiandrosterone sulphate, testosterone, progesterone, corticosterone and aldosterone were also analyzed in relation to adipose-tissue volumetry and distribution and baseline levels of iron-metabolism-related parameters.

## 2. Results

From the initial cohort of 175 participants, we excluded those that had missing or defective MRI scans (n = 26) and those that were subjected to bariatric surgery (n = 18) between baseline and follow-up, to give a final population of 131 subjects at baseline and 79 subjects that agreed to be examined again approximately one year (1.05 ± 0.24 years) later (Figure 1).

Clinical characteristics of the subjects with and without obesity at baseline and the comparison of baseline against follow-up for both groups are shown in Table 1. Body mass index (BMI), SAT and VAT (both raw volumes and corrected indexes) were higher in subjects with obesity, in parallel with decreased glucose-infusion rate during the euglycemic clamp as an indicator of decreased insulin sensitivity. Subjects with obesity also had increased hepatic-iron concentration, circulating ultrasensitive C-reactive protein (CRP), fasting triglycerides, serum uric acid, serum transferrin, and total iron-binding capacity (TIBC).

After stratification by sex and obesity status, we observed the expected differences between men and women with obesity in circulating hepcidin (26.23 ± 16.38 vs. 17.1 ± 10.87 ng/mL, respectively, *p* = 0.046), and ferritin levels (123 (63.5, 251) vs. 63 (35, 94) ng/mL, respectively, *p* = 0.02). Similar differences were observed in subjects without obesity (23.77 ± 9.82 in men vs. 19.50 ± 16.71 ng/mL in women, *p* = 0.035 for serum hepcidin; 196 (133, 235) in men vs. 64 (32, 95) ng/mL in women, *p* < 0.0001 for ferritin).

### 2.1. Cross-Sectional Findings

iVAT was positively correlated with blood hemoglobin, hematocrit, hepatic iron, transferrin, TIBC, ultrasensitive CRP, and uric acid (Figure 2A). Similar associations were found with iSAT regarding hepatic iron, serum transferrin and TIBC, ultrasensitive CRP and uric acid. On the other hand, pSAT was negatively associated with blood hemoglobin, hematocrit, hepatic iron, serum ferritin, hepcidin and serum uric acid.

Baseline insulin sensitivity (M value) measured through euglycemic clamp was negatively correlated with iSAT (r = −0.68, *p* < 0.0001), iVAT (r = −0.74, *p* < 0.0001), blood hematocrit (r = −0.19, *p* = 0.036), hepatic iron (r = −0.34, *p* = 0.001), TIBC (r = −0.22, *p* = 0.017), serum transferrin (r = −0.2, *p* = 0.03), ultrasensitive CRP (r = −0.56, *p* < 0.0001), uric acid (r = −0.50, *p* < 0.0001), while a positive association was found between insulin sensitivity and pSAT (r = 0.23, *p* = 0.011).

Fasting glucose and HbA1c levels did not correlate with haemoglobin, haematocrit, ferritin, transferrin, TIBC, hepcidin, iron and hepatic iron content (all them with r < 0.12 and *p* > 0.1). Interestingly, plasma leptin (a circulating marker of adipose tissue function) was negatively correlated with haemoglobin (r = −0.21, *p* = 0.034), ferritin (r = −0.26, *p* = 0.006) and iron (r = −0.24, *p* = 0.011) and positively with transferrin (r = 0.352, *p* < 0.001) and TIBC (r = 0.353, *p* < 0.001).

### 2.2. Longitudinal Findings

#### 2.2.1. Baseline Parameters of Iron Metabolism and Chronic Inflammation Are Associated with Changes in Adipose Tissue Volumetries

In all subjects, a positive association was found between baseline serum hepcidin and ferritin and changes in iSAT and iVAT (Figure 2B), and negative associations with serum transferrin and TIBC. Serum ferritin and iron were negatively associated with pSAT changes, pointing towards a reduction in pSAT with high baseline ferritin concentrations. Baseline ultrasensitive CRP was positively associated with pSAT changes.

When the findings were stratified by sex, similar associations were found among women (Figure 2C) with serum hepcidin and ferritin being positively associated with changes in iSAT and iVAT. Accordingly, changes in pSAT had a negative relationship with serum ferritin. The positive and negative trends of these associations can be appreciated in Figure 3. In men (Figure 2D), only serum transferrin and TIBC were negatively associated with changes in iSAT and iVAT. Additionally, white blood cell count was reciprocally associated with changes in iVAT (positively) and pSAT (negatively).

According to obesity status, serum hepcidin (positively) and transferrin (negatively) were associated with changes in iSAT and iVAT in subjects without obesity (Figure 2E). In subjects with obesity, iron and ferritin were negatively associated with changes in pSAT (Figure 2F).

#### 2.2.2. Changes in the Chronic-Inflammation Marker, Ultrasensitive CRP, Are Associated with Changes in Adipose Tissue Volumetries

Additionally, after computing the linear correlations between changes in adipose tissue volumetries and changes in iron metabolism (haemoglobin, haematrocrit, iron, transferrin, TIBC and ferritin) or inflammatory parameters (total WBC and ultrasensitive CRP), the only significant associations were found between changes in ultrasensitive CRP and changes in iVAT and pSAT in all subjects (iVAT: r = 0.31, *p* = 0.0084; pSAT: r = −0.27, *p* = 0.018) and women (iVAT: r = 0.35, *p* = 0.011; pSAT: r = −0.28, *p* = 0.047). However, these associations were lost after adjusting for age, sex and BMI in multiple-linear-regression analysis.

#### 2.2.3. Baseline Parameters of Iron Metabolism, Chronic Inflammation and Adipose Tissue Volumetries Are Associated with Changes in Insulin Sensitivity

In all subjects, baseline blood hemoglobin (r = −0.23, *p* = 0.037), haematocrit (r = −0.26, *p* = 0.021), iVAT (r = −0.23, *p* = 0.039) and uric acid (r = −0.31, *p* = 0.004) were negatively associated with changes in insulin sensitivity (M value).

After stratification by obesity status, blood hemoglobin (r = −0.36, *p* = 0.019), haematocrit (r = −0.44, *p* = 0.003) and ultrasensitive CRP (r = −0.37, *p* = 0.016) were significantly associated with the change in insulin sensitivity in subjects without obesity. No significant associations were found in subjects with obesity.

In men, only the associations of ultrasensitive CRP (r = −0.55, *p* = 0.016) and iSAT (r = −0.59, *p* = 0.006) with the change in insulin sensitivity was significant. No significant associations were observed among women.

#### 2.2.4. Adrenal- and Gonadal-Steroid Hormones Are Not Associated with Changes in Adipose Tissue Volumetries or Baseline Parameters of Iron Metabolism

No significant associations between adrenal- and gonadal-steroid hormones and the percent change of iSAT, iVAT and pSAT or baseline levels of serum hepcidin and transferrin were found (Table 2). In addition, while baseline levels of serum ferritin were negatively correlated with estradiol (r = −0.20, *p* = 0.02) and progesterone (r = −0.22, *p* = 0.01), and positively with testosterone (r = 0.38, *p* < 0.001) (Table 2), these associations were lost after adjusting for age, sex and BMI in multiple-linear-regression analysis.

#### 2.2.5. Multiple-Linear-Regression Models

Multiple-linear-regression models were built with the change in adipose tissue volumetries as dependent variables, and age, sex, BMI and baseline serum ferritin, hepcidin, transferrin, ultrasensitive CRP, insulin sensitivity, fasting triglycerides, testosterone and estradiol, as independent variables. The baseline and changes regarding insulin sensitivity after one year were used separately (Table 3). In models with baseline insulin sensitivity, hepcidin (β = 0.283 *p* = 0.05) and insulin sensitivity (β = 0.354, *p* = 0.04) were found to be significantly associated with changes in iVAT, while only hepcidin (β = 0.392, *p* = 0.008) and fasting triglycerides (β = −0.278, *p* = 0.04) were associated with changes in iSAT and pSAT, respectively. In models with changes in insulin sensitivity as the independent variable, only baseline hepcidin was significantly associated with changes in iSAT (β = 0.406, *p* = 0.007) and iVAT (β = 0.306, *p* = 0.04), while changes in insulin sensitivity (β = 0.287, *p* = 0.03) and fasting triglycerides (β = −0.285, *p* = 0.03) were associated with changes in pSAT. Models using changes in ferritin and ultrasensitive CRP instead of the baseline values were also computed, but the results remained essentially the same.

#### 2.2.6. Serum Ferritin and Hepatic Iron

We also explored at baseline the associations with hepatic iron content, as an independent measure of iron stores. As expected, baseline hepatic iron positively correlated with serum ferritin concentration in all subjects (r = 0.28, *p* = 0.0037) and in subjects without obesity (r = 0.42, *p* = 0.0012).

To further examine the potential influence of baseline ferritin and hepatic iron on changes in adipose tissue volumetries, we divided our subjects into low ferritin (n = 66 in baseline, n = 37 in follow-up) and high ferritin (n = 63 in baseline, n = 40 in follow-up), taking the median value for men and women separately as the cut-off points between groups (men: 184 ng/mL; women: 64 ng/mL). A Student’s *t*-test between groups showed increases in iSAT and iVAT with a high baseline ferritin concentration to be statistically significant (iSAT: *p* = 0.008, iVAT: *p* = 0.017). The same analysis was performed separating low (n = 53 in baseline, n = 38 in follow-up) and high (n = 53 in baseline, n = 30 in follow-up) hepatic iron concentration with a cut-off point based on the overall median (11.90 μg/g), but no significant differences were found between those groups.

The sample was later divided into three groups depending on ferritin and hepatic iron: low ferritin and low hepatic iron (n = 28 in baseline, n = 20 in follow-up), high ferritin and high hepatic iron (n = 26 in baseline, n = 16 in follow-up) and low ferritin and high hepatic iron or high ferritin and low hepatic iron (n = 52 in baseline, n = 32 in follow-up). An ANOVA analysis, including a post-hoc multiple comparisons test, showed no significant differences in adiposity changes between these groups. However, comparison between low ferritin/low hepatic iron and high ferritin/high hepatic iron showed significant differences in baseline for the iVAT (*p* = 0.021) and in follow-up for iVAT (*p* = 0.016) and pSAT (*p* = 0.02), pointing towards subjects with higher concentrations of ferritin and hepatic iron having a higher amount and proportion of abdominal visceral adipose tissue. This section may be divided under subheadings. It should provide a concise and precise description of the experimental results and their interpretation, as well as the experimental conclusions that can be drawn.

## 3. Discussion

The results of our study show the association of iron-metabolism markers with adipose tissue volumetries both cross-sectionally and longitudinally. iVAT was positively correlated with circulating iron stores (blood hemoglobin, hematocrit) and markers of tissue iron (hepatic iron, serum hepcidin and ferritin, transferrin and TIBC). Accordingly, pSAT was negatively associated with blood hemoglobin, hematocrit, hepatic iron, serum ferritin and hepcidin, and positively with insulin sensitivity.

Further analysis through multiple-linear-regression models confirmed the independent association of baseline serum hepcidin with iSAT and iVAT changes and fasting triglycerides or changes in insulin sensitivity with pSAT changes.

We also analyzed adrenal- or gonadal-steroid hormones, which are important metabolism modulators that contribute to body-fat distribution [17,18,19] and iron homeostasis [20,21], and no significant association between these hormones and circulating iron metabolism-related parameters or changes in iSAT, iVAT and pSAT were found, suggesting a minor role for these hormones in the relationship between iron and adiposity. However, these data should be considered cautiously, because important aspects such as the reproductive cycle were not taken into account during sampling.

### 3.1. Iron or Chronic Low-Grade Inflammation: The Role of Iron

Through different mechanisms, high hepcidin and high serum-ferritin levels are a sign of an excess iron accumulation inside adipose tissue cells. Hepcidin functions as a regulator of intestinal iron absorption, iron recycling and mobilization from iron stores [22], while serum ferritin is used as a measurement of iron stores, and high values of ferritin are a sign of iron overload [23]. This excess iron has been related to adipocyte hypertrophy and lower adipogenesis [9], which leads to a dysfunction in adipose tissue and an unhealthy fat-mass expansion [24]. In the current study, the negative association between markers of body iron (haemoglobin, ferritin and iron) and circulating leptin, which was in agreement with previous studies [25,26], might reflect the negative impact of iron excess on healthy fat-mass expansion. Supporting this idea, circulating adiponectin, which is an optimal marker of adipose tissue function, was negatively correlated with circulating ferritin and adipocyte iron excess in humans and mice [6,7].

Further inquiry into the role of iron led us to explore the hepatic iron content, finding the expected positive association with adiposity. However, there were no significant differences in changes in adipose tissue between subjects with high or low hepatic iron, nor in subjects with high ferritin/high hepatic iron against low ferritin/low hepatic iron. In the latter case, differences between those groups were found when comparing baseline and follow-up values of adipose tissue separately. The key behind these results could be found in the group of high ferritin/low hepatic iron, as an increase in ferritin concentration without raised hepatic-iron stores might be a sign of inflammatory conditions [27], as discussed below.

Transferrin has also been a factor in the modulation of adipocyte differentiation, as the administration of transferrin improved the attenuation in adipogenesis caused by iron deficiency [9], and in the modulation of iron overload, as increasing serum transferrin reduced tissue iron accumulation [28]. Transferrin saturation, measured as the ratio between serum iron and TIBC, is an indicator of iron exposure. In iron-overload conditions, transferrin saturation is high, and consequently, serum iron is high or TIBC values are low [29]. These notions are in line with our findings. in which negative associations were observed in both transferrin and TIBC with respect to changes in iSAT and iVAT.

### 3.2. The Role of Chronic Low-Grade Inflammation

Higher hepcidin is also known to be associated with inflammatory markers such as, interleukin-6 (IL-6) and CRP [30], leading to lower iron in circulation and difficulties of the proliferation of infectious organisms in the bloodstream [31]. Serum-ferritin levels are also found to be increased during inflammation, and this is known as an inflammatory marker [32]; its involvement could have a protective role, by limiting the production of free radicals and additional pro-inflammatory effects [33].

It is well known that adipose tissue becomes an active metabolic tissue in obesity, leading to alterations in innate immunity and translating to a chronic low-grade inflammatory state [34]. Among those, IL-6 [35] and CRP [36] are up-regulated in parallel with adipose tissue enlargement in subjects with obesity. Current baseline results showed that CRP was very strongly associated with abdominal adipose tissue indexes. In addition to this, previous studies have found that elevated inflammatory markers are associated with weight gain, especially in the case of CRP [15,16].

The associations of hepcidin and ferritin with iVAT and iSAT, respectively, were also found within the group of women. Men typically show increased serum ferritin [37] in parallel with hepcidin [38]. In this context, increased concentrations of hepcidin or ferritin could be the result of expression of a facilitated adipose-tissue expansion in the case of excessive caloric intake, given the well-known requirements of iron for cell replication (in fact, DNA polymerase is an iron-requiring enzyme). Women and subjects without obesity would be more sensitive to these effects. This could be because men and subjects with obesity have already reached a plateau of iron stores, above which the relationships lack linearity. In a state of chronic low-grade inflammation, increases in hepcidin and ferritin would lead to increased iron uptake by adipose tissue cells, an increase in adipose tissue volumetry and to adipocyte dysfunction over time.

### 3.3. The Role of Insulin Sensitivity

Insulin resistance is known to be associated with body iron accumulation [8] and high serum ferritin [39], as well as having a strong relationship with obesity [40], even though some authors have suggested that insulin resistance is the primary driver of inflammation in adipose tissue [41]. Differences in insulin action are known to be associated with redistribution of adipose tissue compartments. We observed a negative association between baseline iVAT and changes in the glucose-infusion rate during the clamp, while the multiple-linear-regression models showed that changes in insulin sensibility were positively associated with changes in pSAT and that baseline insulin sensitivity was negatively associated with changes in iSAT and iVAT. In cross-sectional studies, VAT has been repeatedly identified as the main compartment associated with insulin resistance [42,43,44]. There are relatively few longitudinal studies in the literature. In one report, the subjects with obesity were classified into two groups according to baseline insulin resistance measured using the clamp. In the follow up after 5 years, visceral fat was significantly increased, with the steepest increase found in the obesity insulin-resistant group [45]. Our findings could be seen as confirmatory of these previous ones, but even a short follow-up of one year could point towards a bidirectionality between visceral adiposity and insulin resistance.

The study has its limitations. In some subjects with morbid obesity, subcutaneous tissue was not covered in full during image acquisition, and therefore these volumes were not segmented, which would lead to an underestimation of iSAT values. However, this limitation represents the fact that the findings could be even more significant if the whole SAT abdominal segment could be covered. Contrary to our results, in some studies [1,2,3] serum ferritin was significantly associated with SAT and VAT or increased in subjects with obesity. It is worth noting that we did not include subjects with type 2 diabetes mellitus, and this could influence current results.

Future studies should be designed to investigate whether the early correction of iron excess in humans might prevent the unhealthy expansion of abdominal adipose tissue.

## 4. Materials and Methods

### 4.1. Study Design

A total of 175 participants were recruited consecutively at the Dr. Josep Trueta University Hospital facilities in Girona, Spain from January 2016 to July 2018. From February 2017 to April 2019, 87 of the previously recruited subjects underwent the same protocol as a follow-up assessment. Subjects with obesity were referred from the Endocrinology Department of Dr. Josep Trueta Hospital and from registered research databases, and the subjects without obesity contacted us through e-mail contact or telephone number. This study was approved by the Scientific Research Ethics Committee in September 2015, and all subjects provided informed written consent prior to inclusion. Participants had a first visit where they underwent neuropsychological testing and demographic- and medical-data collection, while MRI acquisitions were performed at a later date. Inclusion criteria were age over 30 years and the ability to understand study procedures. Subjects with BMI ≥ 30 kg/m^2^ were considered for the obesity group and their matched counterparts (BMI from 18.5 to 30 kg/m^2^), according to age and sex, were included in the control group. Exclusion criteria were type 2 diabetes mellitus, NAFLD or liver cirrhosis, major eating or psychiatric-disorder antecedents, anemia or hemoglobinopathy, language disorders, neurological diseases, history of trauma or brain injury, alcohol intake >20 g per day, infection in the last month, serious chronic illness, pregnancy, lactation and MRI contraindications, such as claustrophobia or ferro-magnetic implants. No participants used iron supplements during the follow-up period. Diabetes was excluded, according to the American Diabetes Association criteria, which include fasting glucose equal to or higher than 126 mg/dL and HbA1c equal to or higher than 6.5%.

### 4.2. Clinical and Laboratory Features

Participants underwent anthropometric measurements of height and weight. The body mass index was calculated as weight/height squared (kg/m^2^). Measurements of whole-blood hemoglobin, hematocrit, total white blood cells, iron, ultrasensitive C-reactive protein, transferrin, transferrin saturation (total iron-binding capacity), uric acid, ferritin, glucose, total cholesterol, HDL and LDL cholesterol and triglycerides were performed in the clinical laboratory of the Hospital Dr Josep Trueta in Girona using a routine laboratory test, as detailed elsewhere [4,46]. Circulating hepcidin levels in serum were measured by a solid phase enzyme-linked immunosorbent assay (ELISA) (DRG^®^ Hepcidin 25 (Bioactive) (EIA-5258, DRG International, Inc., Marburg, Germany). Detection limit was 0.35 ng/mL. Intra- and inter-assay coefficients of variation were between 5 and 15%. Serum insulin was measured using a Human Insulin ELISA kit (RIS006R, Biovendor—Laboratorni medicina, a.s., Brno, Czech Republic) with intra- and inter-assay coefficient of variation <7 and <10%, respectively. Insulin resistance was estimated by using the formula of the homeostatic model assessment of insulin resistance (HOMA-IR) as: (Fasting glucose (mg/dL) × Fasting insulin (μIU/mL))/405. Plasma leptin levels were measured by Human Leptin ELISA kit (RAB0333-1KT, Merck Life Science S.L.U., Madrid, Spain). Longitudinal changes were calculated as follows: [(Follow up − baseline)/baseline] × 100.

The determination of adrenal- and gonadal-steroid hormones in serum was carried out using liquid-chromatography with tandem-mass-spectrometry (LC–MS/MS) detection, as previously described [47]. Briefly, sample preparation involved solid-phase extraction in an automated unit from Spark Holland (Emmen, The Netherlands), which was on-line coupled to a LC–MS/MS with a triple quadrupole mass detector from Agilent (Palo Alto, CA, USA). Chromatographic separation of steroids was carried out using a Kinetex C18 analytical column (particle size 2.6 μm, 10 cm length, and 3 mm inner diameter) from Phenomenex (Torrance, CA, USA). The MS/MS detection was carried out in multiple reaction monitoring (MRM) with electrospray ionization in fast-switching polarity mode. Information about the capillary voltage, the nebulizer pressure and parameters for MRM detection are detailed elsewhere [47]. Calibration models were prepared by analysis of aliquots of a serum pool spiked with variable concentrations of the target steroids. The endogenous content of each steroid in the sample loop was subtracted in the preparation of the calibration models. Isotopically labeled steroids were used as internal standards for the quantitative analysis of structurally similar analytes. Serum samples were collected under 8h fasting conditions, between 8.00 and 9.00 a.m., without taking into account the reproductive cycle affecting estradiol and progesterone levels.

Insulin action was determined by the hyperinsulinemic–euglycemic clamp. After an overnight fast, two catheters were inserted into an antecubital vein, one for each arm, used to administer constant infusions of glucose and insulin and to obtain arterialized-venous-blood samples. A 2 h hyperinsulinemic–euglycemic clamp was initiated by a two-step primed infusion of insulin (80 mU/m^2^/min for 5 min, 60 mU/m^2^/min for 5 min) immediately followed by a continuous infusion of insulin at a rate of 40 mU/m^2^/min (regular insulin [Actrapid; Novo Nordisk, Plainsboro, NJ, USA]). Glucose infusion began at min 4 at an initial perfusion rate of 2 mg/kg/min, and it was then adjusted to maintain plasma glucose concentration at 88.3–99.1 mg/dL. Blood samples were collected every 5 min, for the determination of plasma glucose and insulin. Insulin sensitivity was assessed as the mean glucose-infusion rate during the last 40 min. In the stationary equilibrium, the amount of glucose administered (M) equals the glucose taken in by the body tissues, and is a measure of overall insulin sensitivity.

### 4.3. Image Acquisition

The MRI study was performed using a 1.5 Tesla scanner (Ingenia Philips Medical Systems, Eindhoven, The Netherlands) using an 8-channel receiver-coil array. The imaging protocol included three-dimensional volumetric mDIXON gradient-recalled echo acquisitions in the axial plane covering the whole abdominal area, from which fat and water images and in-phase and out-of-phase images were reconstructed (matrix size = 180 × 154, field-of-view = 445 × 381 mm, repetition time (TR) = 5.9 ms, excitation time (TE) = 1.8 and 4 ms, flip angle = 15°, number of slices = 50, acquired voxel volume = 2.5 × 2.5 × 10 mm, reconstructed voxel volume = 2 × 2 × 5 mm). Two stacks were acquired, the first one from the diaphragm to the kidneys, and the second one from the kidneys to below the pubic symphysis. Each stack was acquired within an 11s breath-hold task. R2* values were obtained from the mDIXON sequence. Additional sequences from the protocol include T2* (TR = 120 ms, TE = 14 ms, flip angle = 20°) and proton density (TR = 120 ms, TE = 4 ms, flip angle = 20°).

### 4.4. Image Processing

Hepatic iron content (HIC) was assessed from the R2* values, T2* and proton-density sequences, as described elsewhere [4]. In brief, R2* was calculated as 1/T2* by fitting the monoexponential terms to the T2* signal-decay curve of the respective echo times. R2* values were obtained from the mean averages of signal intensity by drawing three regions of interest (ROI) at the liver. HIC was calculated following the recommendations established by the Spanish Society of Abdominal Diagnostic Imaging (SEDIA). All image analyses were performed by trained and experienced technicians, blinded to clinical information. Abdominal-adipose-tissue volumetric measurements were obtained from the two stacks of mDIXON fat-only images. Scans were converted from DICOM format to NIfTI using the dcm2nii conversion tool (https://people.cas.sc.edu/rorden/mricron/dcm2nii.html, accesses on 8 February 2022) and stitched together using the fslmerge command from the FMRIB Software Library (http://www.fmrib.ox.ac.uk/fsl, accesses on 8 February 2022). The segmentation of volumes of interest (VOI) of SAT and VAT was performed using the software Imfusion Labels (ImFusion GmbH, Munich, Germany). The coverage range was from the superior pole of the left kidney to the pubic symphysis. Percentage of SAT (pSAT) was computed as the ratio of SAT/(VAT+SAT) × 100. SAT and VAT values for each subject were divided by the square of its VOI height (computed as the number of slices of each VOI multiplied by the interslice space of the transverse plane) to minimize the variability due to body morphology. This correction was performed in a similar way as that in which BMI is computed, as the body mass divided by the square of the body height, and the resulting subcutaneous adipose tissue index (iSAT) and visceral adipose tissue index (iVAT) are measured in L/m^2^.

### 4.5. Statistical Analysis

Data was expressed as mean ± standard deviation for normally distributed parameters or median (interquartile range) for non-normally distributed parameters. Normality assumption was checked with the Shapiro–Wilk test. Comparison between two groups were performed with Student’s *t*-test if n > 30 in both groups, otherwise the Wilcoxon rank-sum test was used. ANOVA and post-hoc multiple comparisons tests were applied for comparison between three groups resulting from sex-stratified tertiles of changes in adipose-tissue parameters. Linear correlations between variables were analyzed with Spearman’s rank coefficient. We performed several sets of correlations between (a) baseline adipose-tissue volumetries against iron-metabolism and chronic-inflammation parameters; (b) changes in adipose tissue against baseline iron-metabolism and chronic-inflammation parameters; and (c) changes in insulin sensitivity against baseline adipose-tissue, iron-metabolism and chronic-inflammation parameters. Multiple-linear-regression analysis was performed to observe associations between changes in adipose tissue and other relevant indicators. All statistical analysis was performed with the maximum sample size according to the available data for the variables analyzed. Statistical significance was assumed when the *p* value was <0.05. All the statistical analysis was computed using R version 4.1.2.

## 5. Conclusions

In summary, increased parameters for iron metabolism and chronic inflammation were associated with the redistribution of adipose tissue both cross-sectionally and longitudinally. Baseline serum-hepcidin and serum-ferritin concentrations, as well as low transferrin and TIBC, were the main iron parameters associated with an increase in subcutaneous and visceral adipose tissue after one year of follow up. This outcome could be related to the low-grade inflammation state, characteristic of obesity, and a status of iron overload. These associations were independent of circulating major steroids known to be deregulated in obesity. Under these circumstances, adipocyte hypertrophy and lower adipogenesis triggers an unhealthy expansion of abdominal adipose tissue, while excess abdominal adiposity is associated with an increase in insulin resistance. Therefore, having an iron-overload influence on the expansion of visceral adipose tissue independently of insulin sensitivity suggests that correcting early iron excess could have a preventive effect on insulin-resistance progression associated with visceral adiposity.

## Figures and Tables

**Figure 1 ijms-24-04750-f001:**
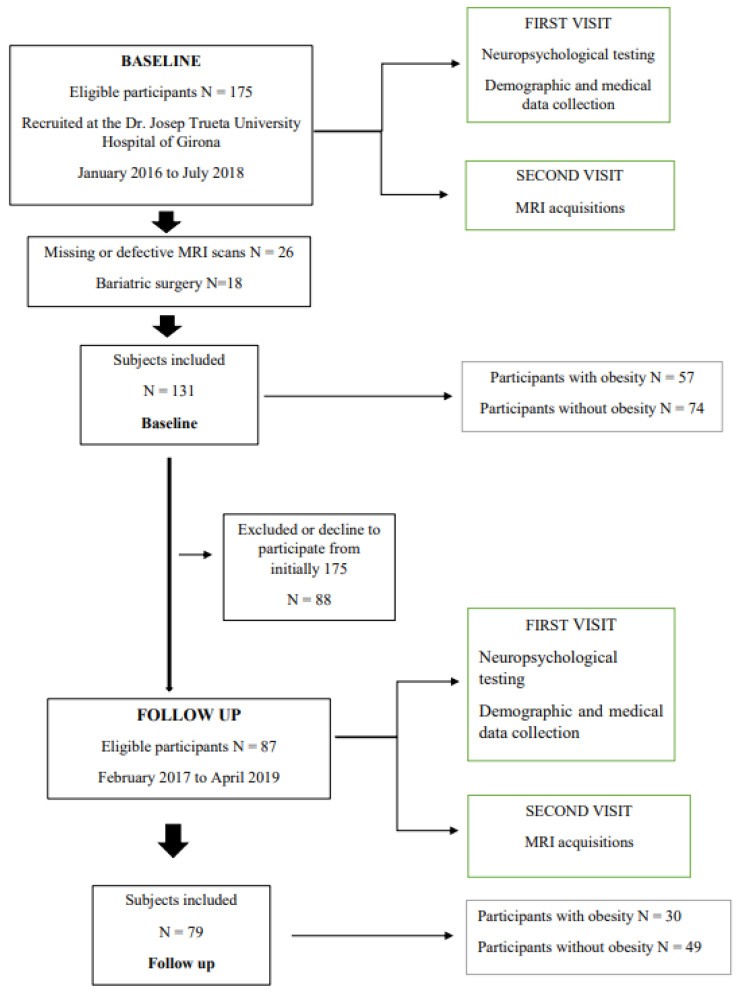
This figure illustrates the flow chart for patient selection.

**Figure 2 ijms-24-04750-f002:**
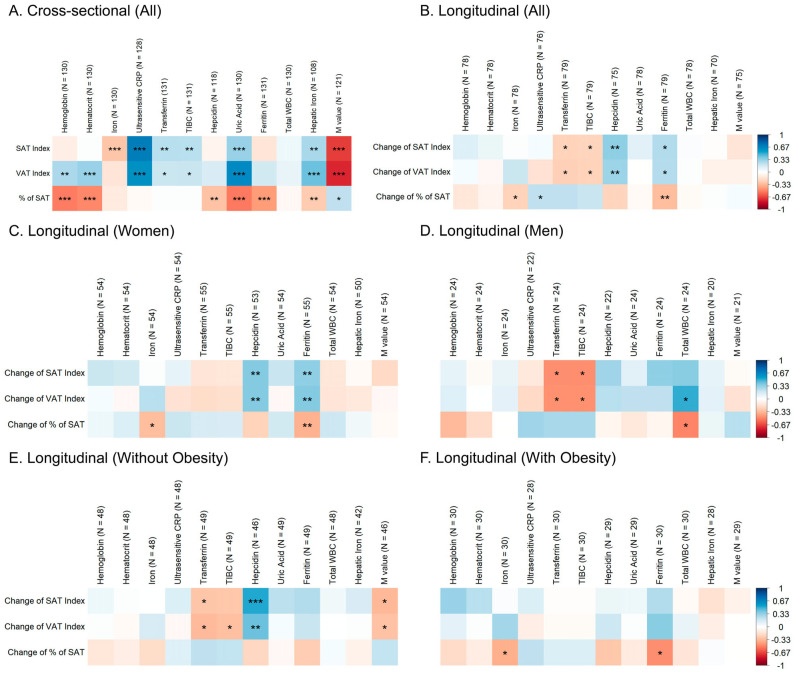
Heatmaps illustrating the Spearman’s rank correlations of (**A**) baseline values of adipose-tissue parameters with baseline iron-metabolism and chronic-inflammation indicators; correlations of changes in adipose-tissue parameters with baseline iron-metabolism and chronic-inflammation indicators in (**B**) all subjects, (**C**) women, (**D**) men, (**E**) subjects without obesity and (**F**) subjects with obesity. * *p* < 0.05, ** *p* < 0.01, *** *p* < 0.001.

**Figure 3 ijms-24-04750-f003:**
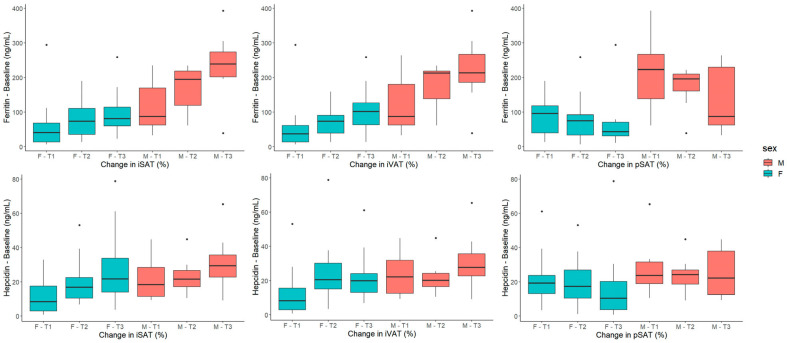
Boxplots showing the relationship between sex-stratified tertiles of changes in adipose-tissue parameters and baseline serum ferritin and hepcidin. Abbreviations: SAT: subcutaneous adipose tissue, VAT: visceral adipose tissue.

**Table 1 ijms-24-04750-t001:** Comparison of baseline and follow-up characteristics of subjects with and without obesity.

	Subjects without Obesity			Subjects with Obesity			*p* *
	Baseline	Follow-Up	*p*	Baseline	Follow-Up	*p*	
N	74	49		57	30		
Clinical							
Age (years)	50.4 (40.4, 58. 8)	54.7 (42.7, 60.2)	<0.001	48.3 (41.8, 54.9)	47.88 ± 10.4	<0.001	0.428
Sex (men/women)	25/49	16/33		16/41	8/22		0.865
Height (cm)	166.3 ± 9.0	165.7 ± 8.4	0.204	162.9 ± 8.4	162.4 ± 7.5	1	0.029
Weight (kg)	66.2 (60.7, 77.4)	67 (60, 78.2)	0.395	112.1 ± 20.8	105. 8 ± 19.5	0.711	<0.001
BMI (kg/m^2^)	24.9 ± 2.7	24.9 ± 2.9	0.261	42. 7 ± 6.4	40.5 ± 7.7	0.703	<0.001
SBP (mmHg)	124.6 ± 16.04	125 (115, 132)	1	138.25 ± 20.13	131.82 ± 9.98	0.414	<0.001
DBP (mmHg)	71.99 ± 10.73	71.71 ± 8.56	0.908	76.76 ± 11.3	73.5 ± 10.5	0.062	0.016
Laboratory							
Hb (g/dL)	13.9 ± 1.3	13.8 ± 1.37	0.594	13.7 ± 1.4	13.8 (13, 14.6)	0.362	0.661
Hct (%)	42.1 ± 3.4	41.2 ± 3.4	0.074	42.47 ± 3.67	42.37 ± 2.86	0.104	0.564
Total WBC (K/μL) ^a^	5.49 (4.87, 6.71)	4.80 (4.08, 6.04)	0.043	6.36 ± 1.96	6.04 (4.92, 6.71)	0.918	0.402
HbA1c (%)	5.4 (5.23, 5.5)	5.47 ± 0.3	0.249	5.6 ± 0.31	5.61 ± 0.27	0.301	0.001
HbA1c (mmol/mol)	35.6 (33.7, 36.7)	36.3 ± 3.3	0.249	37.8 ± 3.4	37.9 ± 2.9	0.351	0.001
Iron (μg/dL)	85 (67, 113)	84 (71, 106)	0.243	77.2 ± 24.6	87 (60.2, 104.2)	0.094	0.008
hsCRP (mg/dL)	0.6 (0.4, 1.6)	0.9 (0.6, 1.7)	0.366	4.2 (2.6, 7.7)	3.1 (1.9, 6.6)	0.198	<0.001
Tf (mg/dL)	246 (224.2, 283. 2)	259.1 ± 36.3	0.522	275.9 ± 36.3	278.5 ± 37.8	0.449	0.001
TIBC (μg/dL)	310.5 (284.8, 352.6)	326.4 ± 41.9	0.621	350.4 ± 46.1	353.6 ± 48.1	0.462	0.001
Uric Acid (mg/dL)	4.49 ± 1.35	4.52 ± 1.32	0.332	5.55 ± 1.2	5.09 ± 0.94	0.509	<0.001
Ferritin (ng/mL)	83 (42, 181.5)	93 (33, 178)	0.274	79 (39, 138)	73 (40.2, 126.2)	0.130	0.601
Hepcidin (ng/mL)	19.1 (10.5, 26.5)			18.0 (9.5, 27.2)			0.623
Hep:Fer ratio	0.2 (0.12, 0.24)			0.2 (0.14, 0.28)			0.845
Glucose (mg/dL)	93 (89, 99)	95 (90, 97.2)	0.894	98.6 ± 12.3	102.2 ± 8.9	0.003	0.061
Insulin (μU/mL)	9.4 ± 5.3	9.5 (8.1, 14.2)	0.001	24.7 ± 11.5	20.3 ± 9.4	0.153	<0.001
M (mg/kg/min)	10.2 ± 3.5	9.6 ± 3.12	0.957	3.7 (2.3, 6.02)	4.09 ± 2.1	0.300	<0.001
HOMA-IR	2.1 (1.3, 3.0)	2.3 (1.8, 3.6)	<0.001	5.5 (3.6, 7.3)	4.4 (3.4, 6.7)	0.319	<0.001
TC (mg/dL)	202.68 ± 40.81	203.08 ± 34.64	0.543	200.25 ± 42.37	196.6 ± 40.94	0.063	0.741
LDL-C (mg/dL)	122.24 ± 34.74	121.37 ± 30.86	0.751	124 (104, 149)	116.7 ± 32.64	0.006	0.364
HDL-C (mg/dL)	61 (51.75, 76.5)	60 (49, 78)	0.315	51.09 ± 12.91	56.17 ± 13.3	0.501	<0.001
TG (mg/dL)	77.5 (58.5, 97.75)	77 (61, 108)	0.680	121 (81, 153)	118.63 ± 57.1	0.299	0.001
Cortisol (ng/mL)	99.0 (51.0, 161.1)			72.6 (33.6, 155.3)			0.7
Estradiol (ng/mL)	0.058 (0.001, 0.112)			0.062 (0.001, 0.115)			0.6
DHEA (ng/mL)	3.09 (1.95, 5.08)			1.61 (0.94, 2.74)			<0.001
Testosterone (ng/mL)	0.14 (0.06, 1.99)			0.11 (0.06, 0.98)			0.04
Progesterone (ng/mL)	0.016 (0.010, 0.031)			0.013 (0.005, 0.023)			0.3
Corticosterone (ng/mL)	0.55 (0.21, 1.86)			0.29 (0.09, 1.17)			0.01
Aldosterone (ng/mL)	0.037 (0.024, 0.054)			0.026 (0.015, 0.038)			0.03
Imaging							
SAT volume (L)	5.82 (4.68, 7.6)	6.35 ± 2	0.030	17.61 ± 4.02	16.64 ± 4.55	0.869	<0.001
VAT volume (L)	1.81 (1.13, 3)	2.12 (1.16, 3.06)	<0.001	5.23 (4.15, 6.36)	5.58 ± 2.18	0.241	<0.001
pSAT (%)	76.9 (68.2, 83.7)	73.4 ± 12.1	0.001	76.5 (72.8, 81.6)	74.3 ± 9.6	0.053	0.636
Height of VOI (m)	0.35 (0.34, 0.37)	0.357 ± 0.019	0.176	0.36 (0.34, 0.37)	0.35 (0.34, 0.37)	0.428	0.456
iSAT (L/m^2^)	49.3 ± 17.3	50.1 ± 16.5	0.122	138.0 ± 29.2	130.7 ± 34.5	0.334	<0.001
iVAT (L/m^2^)	14.0 (8.8, 23.8)	17.6 (9.1, 23.6)	<0.001	39.8 (32.1, 56.4)	38.1 (30.9, 56.6)	0.156	<0.001
Leptin (ng/mL)	1.9 (0.4, 6.1)			23.2 (14.8, 31.7)			<0.001
HIC (μg/g)	10.8 (9.7, 13.1)			12.9 ± 2.31			0.032

Data presented as mean ± standard deviation for normally distributed parameters or median (interquartile range) for non-normally distributed parameters. Baseline and follow-up groups were compared with paired Student’s *t*-test, and the baseline parameters comparison between subjects with and without obesity was analyzed by unpaired Student’s *t*-test. ^a^ Total WBC (K/μL) × 1000. Abbreviations: BMI: body mass index, Hb: haemoglobin, Hct; haematocrit, WBC: white blood cells, HbA1c: glycated hemoglobin, hsCRP: ultrasensitive C-reactive protein, Tf: transferrin, TIBC: total iron-binding capacity, Hep:Fer: Hepcidin:Ferritin, M: M value is a measure of overall insulin sensitivity measured by hyperinsulinemic–euglycemic clamp, HOMA-IR: Homeostasis Model Assessment–Insulin Resistance Index, TC: Total cholesterol, LDL-C: LDL-Cholesterol, HDL-C: HDL-Cholesterol, TG: Triglycerides, SBP: Systolic blood pressure, DBP: Diastolic blood pressure, SAT: subcutaneous abdominal tissue, VOI: volume of interest, VAT: visceral adipose tissue, pSAT: percentage of SAT, iSAT: subcutaneous abdominal tissue index, iVAT: visceral adipose tissue index, HIC: hepatic iron content, *p*: *p* value of the comparison between baseline and follow-up groups, *p* *: *p* value of the comparison between subjects with and without obesity.

**Table 2 ijms-24-04750-t002:** Bivariate correlations between adrenal- and gonadal-steroid hormones and the percent change of iSAT, iVAT and pSAT or baseline levels of serum hepcidin, transferrin and ferritin.

		Changes iSAT	Changes iVAT	Changes pSAT	Hepcidin (ng/mL)	Transferrin (mg/dL)	Ferritin (ng/mL)
Cortisol (ng/mL)	r	−0.08	−0.15	0.12	0.01	−0.07	0.002
	*p*-value	0.43	0.18	0.30	0.91	0.42	0.98
Estradiol (ng/mL)	r	−0.16	−0.02	−0.06	−0.11	0.16	−0.20
	*p*-value	0.15	0.83	0.56	0.22	0.06	0.02
DHEA (ng/mL)	r	−0.14	−0.02	−0.13	−0.12	−0.07	−0.06
	*p*-value	0.19	0.85	0.25	0.17	0.39	0.45
Testosterone (ng/mL)	r	−0.16	−0.07	−0.15	0.14	−0.05	0.38
	*p*-value	0.16	0.51	0.19	0.12	0.59	<0.001
Progesterone (ng/mL)	r	−0.09	0.05	−0.05	−0.13	0.05	−0.22
	*p*-value	0.42	0.63	0.64	0.16	0.59	0.01
Corticosterone (ng/mL)	r	−0.15	−0.15	0.07	−0.08	−0.04	−0.06
	*p*-value	0.18	0.17	0.50	0.4	0.61	0.50
Aldosterone (ng/mL)	r	0.05	0.08	−0.04	−0.10	−0.10	0.02
	*p*-value	0.64	0.47	0.71	0.28	0.24	0.81

**Table 3 ijms-24-04750-t003:** Multiple-linear-regression models of changes in adipose tissue.

	Changes iSAT		Changes iVAT		Changes pSAT	
A	β	*p*-Value	β	*p*-Value	β	*p*-Value
Age (years)	0.004	0.98	−0.089	0.57	0.129	0.42
Sex	0.090	0.81	0.097	0.80	0.225	0.57
BMI (kg/m^2^)	−0.248	0.19	−0.292	0.12	0.207	0.29
Ferritin (ng/mL)	−0.091	0.61	0.010	0.95	−0.076	0.67
Hepcidin (ng/mL)	0.392	0.008	0.283	0.05	−0.064	0.66
Transferrin (mg/dL)	−0.120	0.38	−0.222	0.11	0.206	0.15
hsCRP (mg/dL)	0.011	0.94	−0.196	0.21	0.309	0.06
M value (mg/kg/min)	−0.329	0.06	−0.354	0.04	0.236	0.18
TG (mg/dL)	−0.056	0.66	0.037	0.78	−0.278	0.04
Testosterone (ng/mL)	0.052	0.88	−0.127	0.72	0.414	0.25
Estradiol (ng/mL)	−0.154	0.27	−0.021	0.88	−0.082	0.57
**B**	**β**	***p*-Value**	**β**	***p*-Value**	**β**	***p*-Value**
Age (years)	−0.044	0.78	−0.151	0.35	0.193	0.22
Sex	0.336	0.41	0.359	0.38	−0.087	0.83
BMI (kg/m^2^)	−0.145	0.40	−0.183	0.30	0.180	0.30
Ferritin (ng/mL)	−0.103	0.57	0.004	0.98	−0.102	0.57
Hepcidin (ng/mL)	0.406	0.007	0.306	0.04	−0.066	0.65
Transferrin (mg/dL)	−0.069	0.62	−0.150	0.29	0.140	0.31
hsCRP (mg/dL)	0.136	0.40	−0.060	0.71	0.166	0.31
M value (Change)	−0.222	0.09	−0.218	0.10	0.287	0.03
TG (mg/dL)	−0.021	0.87	0.076	0.56	−0.285	0.03
Testosterone (ng/mL)	0.303	0.41	0.144	0.70	0.118	0.75
Estradiol (ng/mL)	−0.258	0.07	−0.134	0.35	0.025	0.86

Models were built with age, sex, serum hepcidin, serum ferritin, serum transferrin, ultrasensitive C-reactive protein (CRP) and (A) baseline insulin sensitivity or (B) changes in insulin sensitivity. Sample size for both models is 69.

## Data Availability

Not applicable.

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
