# Peer review of "The Longitudinal Changes in Subcutaneous Abdominal Tissue and Visceral Adipose Tissue Volumetries Are Associated with Iron Status"

_ijms, 2023, doi:10.3390/ijms24054750_

Round 1

Reviewer 1 Report (Previous Reviewer 2)

Journal: IJMS    Manuscript ID: ijms-2181947

Authors: Alejandro Hinojosa-Moscoso et al.

Title: “The longitudinal changes in subcutaneous and visceral abdominal adipose tissue volumetries are modulated by iron status”

The authors of the present manuscript explored the potential associations between the markers of iron status and adipose tissue longitudinal changes. According to the authors, iron markers were associated with subcutaneous and visceral adipose tissue changes. The revised version has been improved. However, the following points should be considered.

1.     Please update the methods by adding further information regarding the assessment of the adrenal and gonadal steroid hormones measurements (type of samples, kits, etc.)

2.     According to the authors: “No significant associations between adrenal and gonadal steroid hormones … and ferritin (r<0.13, p>0.2) were found.” Please define if these values apply to all the variables. Moreover, it would be useful if the authors could consider adding the relevant table illustrating these results as supplementary material.

In addition, please add a brief statement to the discussion.

Author Response

Please see also in the attachment the Response to the Editor.

Reviewer 1 comments:

The authors of the present manuscript explored the potential associations between the markers of iron status and adipose tissue longitudinal changes. According to the authors, iron markers were associated with subcutaneous and visceral adipose tissue changes. The revised version has been improved. However, the following points should be considered.

The authors are grateful for the editor and reviewer comments which have contributed to clarify the message of our paper and to improve the quality of our submission. The specific comments are addressed below:

  1. Please update the methods by adding further information regarding the assessment of the adrenal and gonadal steroid hormones measurements (type of samples, kits, etc.)

The following information is now added in methods:

The determination of adrenal and gonadal steroid hormones in serum was carried out by liquid chromatography with tandem mass spectrometry (LC–MS/MS) detection as previously described [45]. Briefly, sample preparation involved solid-phase extraction in an automated unit from Spark Holland (Emmen, The Netherlands), which was on-line coupled to a LC–MS/MS with a triple quadrupole mass detector from Agilent (Palo Alto, USA). Chromatographic separation of steroids was carried out by a Kinetex C18 analytical column (particle size 2.6 μm, 10 cm length, and 3 mm inner diameter) from Phenomenex (Torrance, CA, USA). The MS/MS detection was carried out in multiple reaction monitoring (MRM) with electrospray ionization in fast switching polarity mode. Information about the capillary voltage, the nebulizer pressure and parameters for MRM detection were detailed elsewhere [45]. Calibration models were prepared by analysis of aliquots of a serum pool spiked with variable concentrations of the target steroids. Endogenous content of each steroid in the sample loop was subtracted in the preparation of the calibration models. Isotopically labeled steroids were used as internal standards for quantitative analysis of structurally similar analytes.

  1. Mayneris-Perxachs J, Arnoriaga-Rodríguez M, Luque-Córdoba D, et al. Gut microbiota steroid sexual dimorphism and its impact on gonadal steroids: influences of obesity and menopausal status. Microbiome 2020;8:136.
  2. According to the authors: “No significant associations between adrenal and gonadal steroid hormones … and ferritin (r<0.13, p>0.2) were found.” Please define if these values apply to all the variables. Moreover, it would be useful if the authors could consider adding the relevant table illustrating these results as supplementary material.

To clarify this point, the following information have now been added in results:

Adrenal and gonadal steroid hormones are not associated with changes in adipose tissue volumetries or baseline parameters of iron metabolism

Next, we evaluated whether adrenal and gonadal steroid hormones (cortisol, estradiol, dehydroepiandrosterone sulphate, testosterone, progesterone, corticosterone and aldosterone), known to be deregulated in obesity, were associated with the percent change of iSAT, iVAT and pSAT and with the baseline levels of iron metabolism related parameters. No significant associations between adrenal and gonadal steroid hormones and the percent change of iSAT, iVAT and pSAT or baseline levels of serum hepcidin and transferrin were found (Table 2).  Otherwise, while baseline levels of serum ferritin were negatively correlated with estradiol (r= -0.20, p=0.02) and progesterone (r= -0.22, p=0.01), and positively with testosterone (r= 0.38, p<0.001) (Table 2), these associations were lost after adjusting by age, sex and BMI in multiple linear regression analysis.

Table 2. Bivariate correlations between adrenal and gonaldal steroid hormones and the percent change of iSAT, iVAT and pSAT or baseline levels of serum hepcidin, transferrin and ferritin.

Changes iSAT

Changes iVAT

Changes pSAT

Hepcidin (ng/ml)

Transferrin (mg/dl)

Ferritin (ng/ml)

Cortisol

r

-0.08

-0.15

0.12

0.01

-0.07

0.002

p-value

0.43

0.18

0.30

0.91

0.42

0.98

Estradiol

r

-0.16

-0.02

-0.06

-0.11

0.16

-0.20

p-value

0.15

0.83

0.56

0.22

0.06

0.02

DHEA

r

-0.14

-0.02

-0.13

-0.12

-0.07

-0.06

p-value

0.19

0.85

0.25

0.17

0.39

0.45

Testosterone

r

-0.16

-0.07

-0.15

0.14

-0.05

0.38

p-value

0.16

0.51

0.19

0.12

0.59

<0.001

Progesterone

r

-0.09

0.05

-0.05

-0.13

0.05

-0.22

p-value

0.42

0.63

0.64

0.16

0.59

0.01

Corticosterone

r

-0.15

-0.15

0.07

-0.08

-0.04

-0.06

p-value

0.18

0.17

0.50

0.4

0.61

0.50

Aldosterone

r

0.05

0.08

-0.04

-0.10

-0.10

0.02

p-value

0.64

0.47

0.71

0.28

0.24

0.81

Multiple linear regression models were built with the change in adipose tissue volumetries as dependent variables, and age, sex, BMI and baseline serum ferritin, hepcidin, transferrin, ultrasensitive CRP, insulin sensitivity, fasting triglycerides, testosterone and estradiol as independent variables. Both baseline and changes in insulin sensitivity after one year were used separately (Table 3). In models with baseline insulin sensitivity, hepcidin (β = 0.283 p = 0.05) and insulin sensitivity (β = 0.354, p = 0.04) were found to be significantly associated with changes in iVAT, while only hepcidin (β = 0.392, p = 0.008)  and fasting triglycerides (β = -0.278, p = 0.04) were associated with changes in iSAT and pSAT, respectively. In models with changes in insulin sensitivity as independent variable, only baseline hepcidin was significantly associated with changes in iSAT (β = 0.406, p = 0.007) and iVAT (β = 0.306, p = 0.04), while changes in insulin sensitivity (β = 0.287, p = 0.03) and fasting triglycerides (β = -0.285, p = 0.03) were associated with changes in pSAT. Models using changes in ferritin and ultrasensitive CRP instead of the baseline values were also computed, but the results remained essentially the same.

Table 3. Multiple linear regression models of changes in adipose tissue

Changes iSAT

Changes iVAT

Changes pSAT

A

β

p-value

β

p-value

β

p-value

Age (years)

0.004

0.98

-0.089

0.57

0.129

0.42

Sex

0.090

0.81

0.097

0.80

0.225

0.57

BMI (kg/m2)

-0.248

0.19

-0.292

0.12

0.207

0.29

Ferritin (ng/ml)

-0.091

0.61

0.010

0.95

-0.076

0.67

Hepcidin (ng/ml)

0.392

0.008

0.283

0.05

-0.064

0.66

Transferrin (mg/dl)

-0.120

0.38

-0.222

0.11

0.206

0.15

hsCRP (mg/dl)

0.011

0.94

-0.196

0.21

0.309

0.06

M value (mg/kg/min)

-0.329

0.06

-0.354

0.04

0.236

0.18

TG (mg/dl)

-0.056

0.66

0.037

0.78

-0.278

0.04

Testosterone

0.052

0.88

-0.127

0.72

0.414

0.25

Estradiol

-0.154

0.27

-0.021

0.88

-0.082

0.57

B

β

p-value

β

p-value

β

p-value

Age (years)

-0.044

0.78

-0.151

0.35

0.193

0.22

Sex

0.336

0.41

0.359

0.38

-0.087

0.83

BMI (kg/m2)

-0.145

0.40

-0.183

0.30

0.180

0.30

Ferritin (ng/ml)

-0.103

0.57

0.004

0.98

-0.102

0.57

Hepcidin (ng/ml)

0.406

0.007

0.306

0.04

-0.066

0.65

Transferrin (mg/dl)

-0.069

0.62

-0.150

0.29

0.140

0.31

hsCRP (mg/dl)

0.136

0.40

-0.060

0.71

0.166

0.31

M value (Change)

-0.222

0.09

-0.218

0.10

0.287

0.03

TG (mg/dl)

-0.021

0.87

0.076

0.56

-0.285

0.03

Testosterone

0.303

0.41

0.144

0.70

0.118

0.75

Estradiol

-0.258

0.07

-0.134

0.35

0.025

0.86

In addition, please add a brief statement to the discussion.

To discuss these data, the following text has now been included in discussion:

We also analyzed adrenal or gonadal steroids hormones, which are important metabolism modulators that contribute to body fat distribution [18-20], and no significant association between these hormones and circulating iron metabolism-related parameters or changes in iSAT, iVAT and pSAT were found, suggesting a minor role of these hormones in the relationship between iron and adiposity.

  1. Purnell JQ, Brandon DD, Isabelle LM, Loriaux DL, Samuels MH. Association of 24-hour cortisol production rates, cortisol-binding globulin, and plasma-free cortisol levels with body composition, leptin levels, and aging in adult men and women. J Clin Endocrinol Metab 2004;89:281-287.
  2. Bracht JR, Vieira-Potter VJ, De Souza Santos R, Öz OK, Palmer BF, Clegg DJ. The role of estrogens in the adipose tissue milieu. Ann N Y Acad Sci 2020;1461:127-143.
  3. De Maddalena C, Vodo S, Petroni A, Aloisi AM. Impact of testosterone on body fat composition. J Cell Physiol 2012;227:3744-3748.

Reviewer 2 Report (New Reviewer)

In the present study authors demonstrate the relationship between iron metabolism markers and changes in subcutaneous and visceral adipose tissue in healthy volunteers. The results are of interest and suggest that high iron status is associated with adipose tissue dysfunction, obesity and insulin resistance.

The topic and the results are of interest. I have some comments how the manuscript could be improved.

1)     Were all consecutive obese patients recruited or were they selected from the bigger group? Was the required sample size calculated before the study?

2)     The methods used to measure iron-related parameters and iron-binding proteins should be described in more detail. In particular, limits of quantification as well as intra- and inter-assay CV values should be provided.

3)     Statistical analysis: which 3 groups were compared by ANOVA?

4)     Line 98 and beyond: phrases such as: “SAT and VAT were increased in subjects with obesity” should be better corrected to: :”were higher in subjects with obesity”. “Decreased” or “increased” suggest changes over time whereas in this part of the text authors compare values in both groups at the single time point.

5)     Line 100: is hepatic iron 100 concentration or serum iron concentration mentioned? How was hepatic iron concentration measured? In table 1 only blood iron concentration is presented.

6)     Why sex hormones were taken into consideration in the present study together with iron metabolism scores?

7)     Menopausal status of women should be taken into consideration because menopausal status could significantly affect iron metabolism.

Author Response

Reviewer 2

In the present study authors demonstrate the relationship between iron metabolism markers and changes in subcutaneous and visceral adipose tissue in healthy volunteers. The results are of interest and suggest that high iron status is associated with adipose tissue dysfunction, obesity and insulin resistance.

The topic and the results are of interest. I have some comments how the manuscript could be improved.

The authors are grateful for the reviewer comments which have contributed to clarify the message of our paper and to improve the quality of our submission. The specific comments are addressed below:

  • Were all consecutive obese patients recruited or were they selected from the bigger group?

All consecutive obese patients were recruited following the exclusion and inclusion criteria detailed in methods.

Was the required sample size calculated before the study?

The required sample size was estimated based on our experience in similar studies.

  • The methods used to measure iron-related parameters and iron-binding proteins should be described in more detail. In particular, limits of quantification as well as intra- and inter-assay CV values should be provided.

Thank you. These methods are now described in more detail. Please see below:

“Measurements of whole-blood hemoglobin, hematocrit, total white blood cells, iron, ultrasensitive C-reactive protein, transferrin, transferrin saturation (total iron binding capacity), uric acid, ferritin, glucose, total cholesterol, HDL and LDL cholesterol and triglycerides were performed in the clinical laboratory of the Hospital Dr Josep Trueta in Girona by a routine laboratory test as detailed elsewhere [4, 17]. Circulating hepcidin levels in serum were measured by a solid phase enzyme-linked immunosorbent assay (ELISA) (DRG® Hepcidin 25 (Bioactive) (EIA-5258, DRG International, Inc., Germany). Detection limit was 0.35 ng/mL. Intra- and inter-assay coefficients of variation were between 5 and 15%. Serum insulin was measured by Human Insulin ELISA kit (RIS006R, Biovendor – Laboratorni medicina, a.s., Brno, Czech Republic) with intra- and inter-assay coefficient of variation <7 and <10%, respectively. Insulin resistance was estimated by using the formula of the homeostatic model assessment of insulin resistance (HOMA-IR) as: (Fasting glucose (mg/dl) x Fasting insulin (µIU/ml))/405.”

  • Statistical analysis: which 3 groups were compared by ANOVA?

This was used to evaluate the relationship between sex-stratified tertiles of changes in adipose tissue parameters and baseline serum ferritin and hepcidin, which is shown in Figure 2. This information has now been included in methods. Please see below:

“ANOVA and post-hoc multiple comparisons tests were applied for comparison between three groups resulting from sex-stratified tertiles of changes in adipose tissue parameters”

  • Line 98 and beyond: phrases such as: “SAT and VAT were increased in subjects with obesity” should be better corrected to: ”were higher in subjects with obesity”. “Decreased” or “increased” suggest changes over time whereas in this part of the text authors compare values in both groups at the single time point.

Thank you very much for this appreciation. This sentence is now amended accordingly. Please see below:

BMI, SAT and VAT (both raw volumes and corrected indexes) were higher in subjects with obesity …”

  • Line 100: is hepatic iron concentration or serum iron concentration mentioned? How was hepatic iron concentration measured? In table 1 only blood iron concentration is presented.

It was hepatic iron concentration and appeared in the last row of Table 1 (baseline measurements only). The procedure of this measurement is now better described in methods. Please see below:

“Hepatic iron content (HIC) was assessed from R2* values, T2* and proton density sequences as described elsewhere [4]. In brief, R2* was calculated as 1/T2* by fitting the monoexponential terms to T2* signal decay curve of the respective echo times. R2* values were obtained from the mean averages of signal intensity by drawing three regions of interest (ROI) at liver. HIC was calculated following the recommendations established by the Spanish Society of Abdominal Diagnostic Imaging (SEDIA). All image analyses were performed by trained and experienced technician, blinded to clinical information”

  • Why sex hormones were taken into consideration in the present study together with iron metabolism scores?

Thank you. To clarify this point, the following text has now been included in introduction:

“Considering that adrenal or gonadal steroids hormones are important metabolism modulators that contribute to body fat distribution [18-20] and iron homeostasis [21,22], levels of cortisol, estradiol, dehydroepiandrosterone sulphate, testosterone, progesterone, corticosterone and aldosterone were also analysed in relation to adipose tissue volumetry and distribution and baseline levels of iron metabolism related parameters.”

  1. Hennigar SR, Berryman CE, Harris MN, et al. Testosterone Administration During Energy Deficit Suppresses Hepcidin and Increases Iron Availability for Erythropoiesis. J Clin Endocrinol Metab. 2020;105:dgz316.
  2. Beggs LA, Yarrow JF, Conover CF, et al. Testosterone alters iron metabolism and stimulates red blood cell production independently of dihydrotestosterone. Am J Physiol Endocrinol Metab. 2014;307:E456-E4
  • Menopausal status of women should be taken into consideration because menopausal status could significantly affect iron metabolism.

We fully agree with the reviewer. In order to adjust by menopausal status, in multiple linear regression analysis (Table 3) we included age, sex and estradiol levels.

Reviewer 3 Report (New Reviewer)

Title: The longitudinal changes in subcutaneous and visceral abdominal adipose tissue volumetries are modulated by iron status.

Authors: Alejandro Hinojosa-Moscoso, Anna Motger-Albertí, Elena de la Calle-Vargas, Marian Martí-Navas, Carles Biarnés, María Arnoriaga-Rodríguez, Gerard Blasco, Josep Puig, José María Moreno-Navarrete, José Manuel Fernández-Real.

General comment:

Epidemiological studies suggest that iron deficiency affects 20-40% of obese individuals. The association between obesity and iron deficiency is not fully understood and may be bidirectional. On the one hand, obesity may disrupt iron homeostasis, resulting in iron deficiency. On the other – iron status might determine metabolic health e.g. its cumulation in the liver impacts insulin sensitivity. In their work  Alejandro Hinojosa-Moscoso et al., in a prospective study, investigated whether iron status is linked to changes in adipose tissue distribution in both: obese and normal- to overweight individuals. They found that baseline serum hepcidin and serum ferritin concentrations, low transferrin, and TIBC, were the main iron parameters associated with an increase in subcutaneous and visceral adipose tissue after one year of follow-up. The concept of the study is appealing and the methodology is properly chosen; however, there are some issues regarding the manuscript structure that should be considered before the manuscript is accepted for publication.

Major revisions:

The structure of the manuscript suggests that it has been revised or the Authors changed their concept. The abstract and the introduction do not suggest that the Authors initially aimed to include the results regarding adrenal and gonadal steroids in the analysis (parts of the text marked in red font). Instead of that the Authors write: “Therefore, we aimed to evaluate whether different parameters affected by iron exposure were associated with changes in adiposity and insulin action in a longitudinal study, controlling for inflammatory and oxidative stress markers”. It seems that the initially not gonadal/adrenal steroids but markers of oxidative stress were to be investigated.

Even though the analyses regarding gonadal/adrenal steroids, iron status, and adipose tissue distribution were mainly negative, the mere fact that they were carried out requires a few words of commentary in the introduction and a broader commentary in the discussion. 

However, if the parts of the manuscript written in red correspond to the Authors' revisions, I would appreciate sharing the view of the previous reviewers on the manuscript quality.

Minor revisions:

1)      Abstract: Please include information regarding the statistical significance (P values) in the part of the abstract referring to the description of the results.

2)      Material and Methods: Please provide data if any of the study participants used iron supplements during the follow-up period.

3)      Discussion: Compared to the plethora of results, the discussion is relatively brief. Please consider broader commentaries on the study results including the directions for further research.

Author Response

Reviewer 4 Report (New Reviewer)

In the manuscript entitled “The longitudinal changes in subcutaneous and visceral abdominal adipose tissue volumetries are modulated by iron status,” the author analyzed the correlation between serum iron parameters with longitudinal changes in SAT and VAT. The conclusion of this study can be useful for the related field. I have only some major comments shown below.

1)      It would be better to use a flow chart for patient selection.

2)      The title should be revised as the results are solely based on correlation analysis.

Round 2

Reviewer 1 Report (Previous Reviewer 2)

Journal: IJMS   Manuscript ID: ijms-2181947 (Revised version)

Authors: Alejandro Hinojosa-Moscoso et al.

Title: “The longitudinal changes in subcutaneous and visceral abdominal adipose tissue volumetries are modulated by iron status”

The authors of the present review article have satisfactorily responded to my comments and suggestions and made the necessary changes to the revised paper. Therefore, there are no further comments.

Reviewer 2 Report (New Reviewer)

The manuscript has been revised according to the reviewers' comments.

Reviewer 3 Report (New Reviewer)

I want to express my gratitude for the opportunity to re-review the paper entitled: "The longitudinal changes in subcutaneous and visceral abdominal adipose tissue volumetries are modulated by iron status” by Alejandro Hinojosa-Moscoso et al. Since the authors addressed all my concerns regarding the manuscript concept and structure, I find it acceptable for publishing in the “International Journal of Molecular Sciences”.

This manuscript is a resubmission of an earlier submission. The following is a list of the peer review reports and author responses from that submission.

Round 1

Reviewer 1 Report

In this study, Alejandro and colleagues observed that iron parameters were associated with changes in SAT and VAT. My biggest concern is the insufficient adjustment for potential confounders, which leads to the spurious findings.  This reviewer have some comments and points for clarifications for the authors.

1.      Regarding the study population, were patients with NAFLD or liver cirrhosis included?

2.      Line 392, the MR study should be MRI study.

3.      Please give details of the assessment of exposures, which is very important for others to evaluate whether the method is reliable.  It could be in supplemental methods if the information is too long.

4.      Can the authors comment on the selection of the confounders? The current model seems to be very parsimonious, without adjustment for baseline BMI, diet, physical activity, sleep, lipids profile, blood pressure parameters…

Author Response

Manuscript Number: ijms-2063787 entitled The longitudinal changes in subcutaneous and visceral abdominal adipose tissue volumetries are modulated by iron status

The authors are grateful for the editor and reviewer comments which have contributed to clarify the message of our paper and to improve the quality of our submission. The specific comments are addressed below:

Reviewer 1

In this study, Alejandro and colleagues observed that iron parameters were associated with changes in SAT and VAT. My biggest concern is the insufficient adjustment for potential confounders, which leads to the spurious findings.  This reviewer have some comments and points for clarifications for the authors.

Thank you very much for your constructive comments.

  1. Regarding the study population, were patients with NAFLD or liver cirrhosis included?

Thank you for this appreciation. Patients with NAFLD or liver cirrhosis were excluded. This information is now specified in methods. Please see below:

Methods

“Exclusion criteria were type 2 diabetes mellitus, impaired glucose tolerance, NAFLD or liver cirrhosis, major eating or psychiatric disorder antecedents, anemia or hemoglobinopathy, language disorders, neurological diseases, history of trauma or brain injury, alcohol intake > 20g per day, infection in the last month, serious chronic illness, pregnancy, lactation and MRI contraindications, such as claustrophobia or ferro-magnetic implants.”

  1. Line 392, the MR study should be MRI study.

Thank you. This is now amended accordingly.

  1. Please give details of the assessment of exposures, which is very important for others to evaluate whether the method is reliable.  It could be in supplemental methods if the information is too long.

The exclusion criteria took into account the main factors influencing serum ferritin. For instance, the exclusion criteria regarding alcohol intake > 20 g per day was conceived to minimize the possible impact of alcohol intake on serum ferritin levels.

  1. Can the authors comment on the selection of the confounders? The current model seems to be very parsimonious, without adjustment for baseline BMI, diet, physical activity, sleep, lipids profile, blood pressure parameters…

The selection of the confounders was performed considering those baseline iron metabolism-related parameters that were associated with longitudinal changes in iSAT, iVAT and pSAT (please see Figure 1B).  New multiple linear regression analysis including BMI and fasting triglycerides have now been performed and included in new Table 4 (now Table 2), and results and discussion section has now been modified accordingly. Please see below:

Results

Multivariant Multiple linear regression models were built with the change in adipose tissue volumetries as dependent variables, and age, sex, BMI and baseline serum ferritin, hepcidin, transferrin, ultrasensitive CRP, insulin sensitivity and fasting triglycerides as independent variables. Both baseline and changes in insulin sensitivity after one year were used separately (Table 2). In models with baseline insulin sensitivity, hepcidin (β = 0.370, p = 0.011) and insulin sensitivity (β = -0.371, p = 0.029) were found to be the single parameter significantly associated with changes in iSAT (β = 0.352, p = 0.013), while only insulin sensitivity (β = -0.353, p = 0.037) and fasting triglycerides (β = -0.282, p = 0.037) were associated with iVAT (β = -0.291, p = 0.04) and pSAT (β = 0.315, p = 0.031), respectively. In models with changes in insulin sensitivity as independent variable, only baseline hepcidin was significantly associated with changes in iSAT (β = 0.386, p = 0.010) and iVAT (β = 0.295, p = 0.047), while changes in insulin sensitivity (β = 0.294, p = 0.019) and fasting triglycerides (β = -0.283, p = 0.028) were associated with changes in pSAT (β = 0.327, p = 0.014).

Discussion

“Further analysis through multivariant multiple linear regression models confirmed the independent association of baseline serum hepcidin with iSAT and iVAT changes and fasting triglycerides or changes in insulin sensitivity with pSAT changes.”

After controlling for these confounders, the associations with CRP were no longer observed. For this reason, we deleted this text:

“It is well known that adipose tissue becomes an active metabolic tissue in obesity, leading to alterations in innate immunity and translating it to a chronic low-grade inflammatory state [28]. Among those, interleukin-6 (IL-6) [29] and CRP [30] are up-regulated in parallel to adipose tissue enlargement in subjects with obesity. Current baseline results showed that CRP was very strongly associated with abdominal adipose tissue indexes. In addition to that, previous studies have found that elevated inflammatory markers are associated with weight gain, especially in the case of CRP [15, 16]. Our multivariant regression models showed opposite results: baseline CRP values were associated with a reduction of iVAT and increase of pSAT, in line with a longitudinal study from Finland [31]. However, we also found a positive correlation between changes in CRP and changes in iVAT and a negative relationship with changes in pSAT, which could point to baseline CRP not being able to predict changes in abdominal adipose tissue because inflammation might go in parallel to obesity

“We observed a negative association between baseline iVAT and changes in the glucose infusion rate during the clamp, while the multivariate multiple linear regression models showed that changes in insulin sensibility were positively associated with changes in pSAT and that baseline insulin sensitivity was negatively associated with changes in iSAT and iVAT.”

Table 2:

Changes iSAT

Changes iVAT

Changes pSAT

A

β

p-value

β

p-value

β

p-value

Age

0.082

0.555

-0.087

0.542

0.186

0.196

Sex

0.044

0.784

0.223

0.168

-0.180

0.282

BMI

-0.226

0.226

-0.283

0.130

0.200

0.299

Ferritin

-0.058

0.737

0.025

0.882

-0.089

0.615

Hepcidin

0.370

0.011

0.279

0.052

-0.072

0.621

Transferrin

-0.122

0.373

-0.218

0.114

0.191

0.180

CRP

-0.032

0.827

-0.203

0.175

0.287

0.065

M value (Baseline)

-0.371

0.029

-0.353

0.037

0.194

0.260

Fasting triglycerides

-0.064

0.617

0.035

0.783

-0.282

0.037

B

β

p-value

β

p-value

β

p-value

Age

0.094

0.510

-0.080

0.571

0.185

0.184

Sex

0.027

0.873

0.211

0.206

-0.205

0.209

BMI

-0.055

0.744

-0.136

0.419

0.172

0.297

Ferritin

-0.085

0.633

0.015

0.933

-0.118

0.494

Hepcidin

0.386

0.010

0.295

0.047

-0.061

0.671

Transferrin

-0.074

0.596

-0.152

0.277

0.135

0.322

CRP

0.052

0.739

-0.103

0.510

0.171

0.267

M value (Change)

-0.158

0.214

-0.185

0.143

0.294

0.019

Fasting triglycerides

-0.008

0.948

0.082

0.525

-0.283

0.028

Reviewer 2 Report

Journal: IJMS                                                                           
Manuscript ID: ijms-2063787

Authors: Alejandro Hinojosa-Moscoso et al.

Title: “The longitudinal changes in subcutaneous and visceral abdominal adipose tissue volumetries are modulated by iron status”

Comments to the authors:

1.     How did the authors exclude the presence of diabetes or prediabetes?

2.     Did the authors exclude participants with anemia or hemoglobinopathy?

3.     Since the authors used the Shapiro-Wilk test to examine the normality of the variables, I would suggest using parametric or non-parametric tests based on the relevant findings (T-test/Wilcoxon, ANOVA/Kruskal-Wallis, pearson/spearman’s, etc.). Please specify the test(s) used for the multiple comparison tests.

4.     Please add in the tables the p-value for sex.

5.     The data presented in table 1 are included in Tables 2 and 3. Therefore, tables 1,2,3 could be merged and avoid presenting the same data twice.

6.     It would be useful to clarify/confirm if the linear regression models are multivariant or multivariable. I would suggest performing the linear regression models using the (delta) changes of the examined variables. Moreover, according to the authors, “Changes in iron metabolism and chronic inflammation parameters are associated with changes in adipose tissue volumetries”. However, which parameters were included in the models and the dependent variables are not clear. Could the authors clarify this point further?

7.     Almost all the analyses are performed based on the baseline values. At the same time, it would also be interesting to explore the impact of the change in these parameters on the outcomes of interest.

8.     The authors should specify in the manuscript whether the “change” in the examined variables corresponds to delta, fold, etc.

9.     Could the authors clarify what the M value in the tables means?

Author Response

Reviewer 2

Manuscript Number: ijms-2063787 entitled The longitudinal changes in subcutaneous and visceral abdominal adipose tissue volumetries are modulated by iron status

The authors are grateful for the editor and reviewer comments which have contributed to clarify the message of our paper and to improve the quality of our submission. The specific comments are addressed below:

Comments to the authors:

  1. How did the authors exclude the presence of diabetes or prediabetes?

The presence of diabetes, but not prediabetes, was excluded according to American Diabetes Association criteria using fasting glucose and HbA1c parameters. Participants were excluded when fasting glucose were equal to or higher than 126 mg/dl and HbA1c equal to or higher than 6.5%.

The following information has now been included in methods:

Exclusion criteria were type 2 diabetes mellitus, impaired glucose tolerance, NAFLD or liver cirrhosis, major eating or psychiatric disorder antecedents, anemia or hemoglobinopathy, language disorders, neurological diseases, history of trauma or brain injury, alcohol intake > 20g per day, infection in the last month, serious chronic illness, pregnancy, lactation and MRI contraindications, such as claustrophobia or ferro-magnetic implants. Diabetes was excluded according to American Diabetes Association criteria, which include fasting glucose equal to or higher than 126 mg/dl and HbA1c equal to or higher than 6.5%.

  1. Did the authors exclude participants with anemia or hemoglobinopathy?

Anemia or hemoglobinopathy were excluded assessing haemoglobin and haematocrit. Both haemoglobin and haematocrit levels were in the normal range. This information is now included in methods. Please see below:

Exclusion criteria were type 2 diabetes mellitus, impaired glucose tolerance, NAFLD or liver cirrhosis, major eating or psychiatric disorder antecedents, anemia or hemoglobinopathy, language disorders, neurological diseases, history of trauma or brain injury, alcohol intake > 20g per day, infection in the last month, serious chronic illness, pregnancy, lactation and MRI contraindications, such as claustrophobia or ferro-magnetic implants. Diabetes was excluded according to American Diabetes Association criteria, which include fasting glucose equal to or higher than 126 mg/dl and HbA1c equal to or higher than 6.5%.

  1. Since the authors used the Shapiro-Wilk test to examine the normality of the variables, I would suggest using parametric or non-parametric tests based on the relevant findings (T-test/Wilcoxon, ANOVA/Kruskal-Wallis, pearson/spearman’s, etc.). Please specify the test(s) used for the multiple comparison tests.

We fully agree with the reviewer. Comparison between two groups were performed with T-test/Wilcoxon according to the normality of the variables. Since the most of iron metabolism or inflammatory variables were non-normally distributed parameters, for bivariate correlations Spearman’s rank coefficient was used. For the multiple comparison tests, multiple linear regression analyses were performed. This information was described in methods. Please see below:

Statistical analysis

Data was expressed as mean ± standard deviation for normally distributed parameters or median (interquartile range) for non-normally distributed parameters. Normality assumption was checked with the Shapiro-Wilk test. Comparison between two groups were performed with Student’s t-test if n > 30 in both groups, otherwise Wilcoxon rank-sum test was used. ANOVA and post-hoc multiple comparisons tests were applied for comparison between three groups. Linear correlations between variables were analysed with Spearman’s rank coefficient. We performed several sets of correlations between a) baseline adipose tissue volumetries against iron metabolism and chronic inflammation parameters; b) changes in adipose tissue against baseline iron metabolism and chronic inflammation parameters; and c) changes in insulin sensitivity against baseline adipose tissue, iron metabolism and chronic inflammation parameters. Multivariant Multiple linear regression analysis was performed to observe associations between changes in adipose tissue and other relevant indicators. All statistical analysis was performed with the maximum sample size according to the available data for the variables analyzed. Statistical significance was assumed when the P value was < 0.05. All the statistical analysis was computed using R version 4.1.2.

  1. Please add in the tables the p-value for sex.
  2. p-value for sex were added in Table 1. Please see below the new Table 1.
  3. The data presented in table 1 are included in Tables 2 and 3. Therefore, tables 1,2,3 could be merged and avoid presenting the same data twice.

We fully agree with the reviewer. Tables 1,2,3 have now been merged to avoid presenting the same data twice. Please see new Table 1.

New Table 1

Subjects without obesity

Subjects with obesity

p *

Baseline

Follow-up

p

Baseline

Follow-up

p

N

74

49

57

30

Clinical

Age (years)

50.4 (40.4, 58. 8)

54.7 (42.7, 60.2)

< 0.001

48.3 (41.8, 54.9)

47.88 ± 10.4

< 0.001

0.428

Sex (men/women)

25 / 49

16 / 33

16 / 41

8 / 22

0.865

Height (cm)

166.3 ± 9.0

165.7 ± 8.4

0.204

162.9 ± 8.4

162.4 ± 7.5

1

0.029

Weight (kg)

66.2 (60.7, 77.4)

67 (60, 78.2)

0.395

112.1 ± 20.8

105. 8 ± 19.5

0.711

< 0.001

BMI (kg/m2)

24.9 ± 2.7

24.9 ± 2.9

0.261

42. 7 ± 6.4

40.5 ± 7.7

0.703

< 0.001

Laboratory

Hb (g/dl)

13.9 ± 1.3

13.8 ± 1.37

0.594

13.7 ± 1.4

13.8 (13, 14.6)

0.362

0.661

Hct (%)

42.1 ± 3.4

41.2 ± 3.4

0.074

42.47 ± 3.67

42.37 ± 2.86

0.104

0.564

Total WBC (K/µl)a

5.49 (4.87, 6.71)

4.80 (4.08, 6.04)

0.043

6.36 ± 1.96

6.04 (4.92, 6.71)

0.918

0.402

HbA1c (%)

5.4 (5.23, 5.5)

5.47 ± 0.3

0.249

5.6 ± 0.31

5.61 ± 0.27

0.301

0.001

HbA1c (mmol/mol)

35.6 (33.7, 36.7)

36.3 ± 3.3

0.249

37.8 ± 3.4

37.9 ± 2.9

0.351

0.001

Iron (µg/dl)

85 (67, 113)

84 (71, 106)

0.243

77.2 ± 24.6

87 (60.2, 104.2)

0.094

0.008

hsCRP (mg/dl)

0.6 (0.4, 1.6)

0.9 (0.6, 1.7)

0.366

4.2 (2.6, 7.7)

3.1 (1.9, 6.6)

0.198

< 0.001

Tf (mg/dl)

246 (224.2, 283. 2)

259.1 ± 36.3

0.522

275.9 ± 36.3

278.5 ± 37.8

0.449

0.001

TIBC (µg/dl)

310.5 (284.8, 352.6)

326.4 ± 41.9

0.621

350.4 ± 46.1

353.6 ± 48.1

0.462

0.001

Uric Acid (mg/dl)

4.49 ± 1.35

4.52 ± 1.32

0.332

5.55 ± 1.2

5.09 ± 0.94

0.509

< 0.001

Ferritin (ng/ml)

83 (42, 181.5)

93 (33, 178)

0.274

79 (39, 138)

73 (40.2, 126.2)

0.130

0.601

Hepcidin (ng/ml)

19.1 (10.5, 26.5)

18.0 (9.5, 27.2)

0.623

Hep:Fer ratio

0.2 (0.12, 0.24)

0.2 (0.14, 0.28)

0.845

Glucose (mg/dl)

93 (89, 99)

95 (90, 97.2)

0.894

98.6 ± 12.3

102.2 ± 8.9

0.003

0.061

Insulin (µU/ml)

9.4 ± 5.3

9.5 (8.1, 14.2)

0.001

24.7 ± 11.5

20.3 ± 9.4

0.153

< 0.001

M (mg/kg/min)

10.2 ± 3.5

9.6 ± 3.12

0.957

3.7 (2.3, 6.02)

4.09 ± 2.1

0.300

< 0.001

HOMA-IR

2.1 (1.3, 3.0)

2.3 (1.8, 3.6)

< 0.001

5.5 (3.6, 7.3)

4.4 (3.4, 6.7)

0.319

< 0.001

TC (mg/dl)

202.68 ± 40.81

203.08 ± 34.64

0.543

200.25 ± 42.37

196.6 ± 40.94

0.063

0.741

LDL-C (mg/dl)

122.24 ± 34.74

121.37 ± 30.86

0.751

124 (104, 149)

116.7 ± 32.64

0.006

0.364

HDL-C (mg/dl)

61 (51.75, 76.5)

60 (49, 78)

0.315

51.09 ± 12.91

56.17 ± 13.3

0.501

< 0.001

TG (mg/dl)

77.5 (58.5, 97.75)

77 (61, 108)

0.680

121 (81, 153)

118.63 ± 57.1

0.299

0.001

SBP (mmHg)

124.6 ± 16.04

125 (115, 132)

1

138.25 ± 20.13

131.82 ± 9.98

0.414

< 0.001

DBP (mmHg)

71.99 ± 10.73

71.71 ± 8.56

0.908

76.76 ± 11.3

73.5 ± 10.5

0.062

0.016

Imaging

SAT volume (l)

5.82 (4.68, 7.6)

6.35 ± 2

0.030

17.61 ± 4.02

16.64 ± 4.55

0.869

< 0.001

VAT volume (l)

1.81 (1.13, 3)

2.12 (1.16, 3.06)

< 0.001

5.23 (4.15, 6.36)

5.58 ± 2.18

0.241

< 0.001

pSAT (%)

76.9 (68.2, 83.7)

73.4 ± 12.1

0.001

76.5 (72.8, 81.6)

74.3 ± 9.6

0.053

0.636

Height of VOI (m)

0.35 (0.34, 0.37)

0.357 ± 0.019

0.176

0.36 (0.34, 0.37)

0.35 (0.34, 0.37)

0.428

0.456

iSAT (l/m2)

49.3 ± 17.3

50.1 ± 16.5

0.122

138.0 ± 29.2

130.7 ± 34.5

0.334

< 0.001

iVAT (l/m2)

14.0 (8.8, 23.8)

17.6 (9.1, 23.6)

< 0.001

39.8 (32.1, 56.4)

38.1 (30.9, 56.6)

0.156

< 0.001

Leptin (ng/ml)

1.9 (0.4, 6.1)

23.2 (14.8, 31.7)

< 0.001

HI (µg/g)

10.8 (9.7, 13.1)

12.9 ± 2.31

0.032

Data presented as mean ± standard deviation for normally distributed parameters or median (interquartile range) for non-normally distributed parameters. Baseline and follow-up groups were compared with paired Student's t-test, and the baseline parameters comparison between subjects with and without obesity was analysed by unpaired Student's t-test. aTotal WBC (K/µl)x1000.

Abbreviations: BMI: body mass index, Hb: haemoglobin, Hct; haematocrit, WBC: white blood cells, HbA1c: glycated hemoglobin, hsCRP: ultrasensitive C-reactive protein, Tf: transferrin, TIBC: total iron binding capacity, Hep:Fer: Hepcidin:Ferritin, M: M value is a measure of overall insulin sensitivity measured by hyperinsulinemic–euglycemic clamp, HOMA-IR: Homeostasis Model Assessment–Insulin Resistance Index, TC: Total cholesterol, LDL-C: LDL-Cholesterol, HDL-C: HDL-Cholesterol, TG: Triglycerides, SBP: Systolic Blood pressure, DBP: Diastolic blood pressure, SAT: subcutaneous adipose tissue, VOI: volume of interest, VAT: visceral adipose tissue, pSAT: percentage of SAT, iSAT: subcutaneous adipose tissue index, iVAT: visceral adipose tissue index, HI: hepatic iron, p: p value of the comparison between baseline and follow-up groups, p*: p value of the comparison between subjects with and without obesity.

  1. - It would be useful to clarify/confirm if the linear regression models are multivariant or multivariable.

We have now changed “multivariant” into “multiple” throughout the manuscript to clarify that multiple linear regression analysis was performed.

  • I would suggest performing the linear regression models using the (delta) changes of the examined variables.

We have now performed multiple linear regression analysis for ultrasensitive CRP changes, which is the only one correlated with changes in iVAT and pSAT. This new analysis is now added in results. Please see below:

Results

“the only significant associations were found between changes in ultrasensitive CRP and changes in iVAT and pSAT in all subjects (iVAT: r = 0.31, p = 0.0084; pSAT: r = -0.27, p = 0.018) and women (iVAT: r = 0.35, p = 0.011; pSAT: r = -0.28, p = 0.047). However, these associations were lost after adjusting by age, sex and BMI in multiple linear regression analysis.”

  • Moreover, according to the authors, “Changes in iron metabolism and chronic inflammation parameters are associated with changes in adipose tissue volumetries”. However, which parameters were included in the models and the dependent variables are not clear. Could the authors clarify this point further?

Thank you for this appreciation. To clarify this point, the following changes have now been performed in this paragraph:

Changes in iron metabolism and chronic inflammation parameters the chronic inflammation marker ultrasensitive CRP are associated with changes in adipose tissue volumetries

Additionally, after computing the linear correlations between changes in adipose tissue volumetries and changes in iron metabolism (haemoglobin, haematrocrit, iron, transferrin, TIBC and ferritin) or inflammatory parameters (total WBC and ultrasensitive CRP), …”

  1. Almost all the analyses are performed based on the baseline values. At the same time, it would also be interesting to explore the impact of the change in these parameters on the outcomes of interest.

We fully agree with the reviewer. Changes in some markers of iron metabolism and inflammation were analyzed, but only changes in ultrasensitive CRP were significantly correlated with changes in iVAT and pSAT, but these associations were lost after adjusting by age, sex and BMI in multiple linear regression analysis. This information is now detailed in results. Please see below:

Additionally, after computing the linear correlations between changes in adipose tissue volumetries and changes in iron metabolism (haemoglobin, haematrocrit, iron, transferrin, TIBC and ferritin) or inflammatory parameters (total WBC and ultrasensitive CRP), the only significant associations were found between changes in ultrasensitive CRP and changes in iVAT and pSAT in all subjects (iVAT: r = 0.31, p = 0.0084; pSAT: r = -0.27, p = 0.018) and women (iVAT: r = 0.35, p = 0.011; pSAT: r = -0.28, p = 0.047). However, these associations were lost after adjusting by age, sex and BMI in multiple linear regression analysis.”

  1. The authors should specify in the manuscript whether the “change” in the examined variables corresponds to delta, fold, etc.

This point is now specified in methods. Please see below:

Methods

Longitudinal changes were calculated as follows: [(Follow up– baseline)/baseline] *100

  1. Could the authors clarify what the M value in the tables means?

This is now clarified in the footnote of Table 1. In addition, the procedure of insulin sensitivity has now been included in methods. Please see below:

In the footnote of Table 1

M: M value is a measure of overall insulin sensitivity measured by hyperinsulinemic–euglycemic clamp

In methods

Insulin action was determined by the hyperinsulinemic–euglycemic clamp. After an overnight fast, two catheters were inserted into an antecubital vein, one for each arm, used to administer constant infusions of glucose and insulin and to obtain arterialized venous blood samples. A 2-h hyperinsulinemic–euglycemic clamp was initiated by a two-step primed infusion of insulin (80 mU/m2/min for 5 min, 60 mU/m2/min for 5 min) immediately followed by a continuous infusion of insulin at a rate of 40 mU/m2/min (regular insulin [Actrapid; Novo Nordisk, Plainsboro, NJ]). Glucose infusion began at min 4 at an initial perfusion rate of 2 mg/kg/min, and it was then adjusted to maintain plasma glucose concentration at 88.3–99.1 mg/dL. Blood samples were collected every 5 min for determination of plasma glucose and insulin. Insulin sensitivity was assessed as the mean glucose infusion rate during the last 40 min. In the stationary equilibrium, the amount of glucose administered (M) equals the glucose taken by the body tissues and is a measure of overall insulin sensitivity.

Reviewer 3 Report

Dear Editor,

The manuscript entitled " The longitudinal changes in subcutaneous and visceral abdominal adipose tissue volumetries are modulated by iron status" is an original manuscript reporting the association between adipose tissue changes and markers of iron metabolism.  The association between iron and adipose tissue physiology has been intensely investigated in recent years, with hundreds of published articles. In particular, the relationship between obesity and markers of iron metabolism has also been shown and there are already review manuscripts on the topic. In this regard, adipose tissue macrophages seem to have an important role in iron regulation and some data support that iron is important for brown adipose tissue activity.

The study has some merits, but the authors could have better stratified the patients, namely according to their glucose metabolism. Some of the patients apparently are at least prediabetic and this could have been addressed in data analysis. Moreover, only looking at obesity is very limiting. Also, measuring some markers of adipose tissue function could improve the quality of the manuscript. In its current form, I believe the manuscript does not add anything new and does not have enough quality to be published in IJMS.

Author Response

Reviewer 3

Manuscript Number: ijms-2063787 entitled The longitudinal changes in subcutaneous and visceral abdominal adipose tissue volumetries are modulated by iron status

The authors are grateful for the editor and reviewer comments which have contributed to clarify the message of our paper and to improve the quality of our submission. The specific comments are addressed below:

The manuscript entitled " The longitudinal changes in subcutaneous and visceral abdominal adipose tissue volumetries are modulated by iron status" is an original manuscript reporting the association between adipose tissue changes and markers of iron metabolism.  The association between iron and adipose tissue physiology has been intensely investigated in recent years, with hundreds of published articles. In particular, the relationship between obesity and markers of iron metabolism has also been shown and there are already review manuscripts on the topic. In this regard, adipose tissue macrophages seem to have an important role in iron regulation and some data support that iron is important for brown adipose tissue activity.

The study has some merits, but the authors could have better stratified the patients, namely according to their glucose metabolism. Some of the patients apparently are at least prediabetic and this could have been addressed in data analysis. Moreover, only looking at obesity is very limiting. Also, measuring some markers of adipose tissue function could improve the quality of the manuscript. In its current form, I believe the manuscript does not add anything new and does not have enough quality to be published in IJMS.

Thank you very much for your constructive comments. To evaluate the possible contribution of prediabetes and some marker of adipose tissue function, bivariate correlations between fasting glucose, HbA1c and circulating leptin and iron metabolism parameters have now been analysed. These new data have now been added in results and discussion. Please see below:

In methods

Plasma leptin levels were measured by Human Leptin ELISA kit (RAB0333-1KT, Sigma-Aldrich). 

In results

Fasting glucose and HbA1c levels did not correlate with haemoglobin, haematocrit, ferritin, transferrin, TIBC, hepcidin, iron and hepatic iron content (all them with r<0.12 and p>0.1). Interestingly, plasma leptin (a circulating marker of adipose tissue function) was negatively correlated with haemoglobin (r=-0.21, p=0.034), ferritin (r=-0.26, p=0.006) and iron (r=-0.24, p=0.011) and positively with transferrin (r=0.352, p<0.001) and TIBC (r=0.353, p<0.001). 

In discussion

Through different mechanisms, high hepcidin and high serum ferritin levels are a sign of an excess iron accumulation inside adipose tissue cells. Hepcidin functions as a regulator of intestinal iron absorption, iron recycling and mobilization from iron stores [18], while serum ferritin is used as a measurement of iron stores and high values of ferritin are a sign of iron overload [19]. This excess iron has been related to adipocyte hypertrophy and lower adipogenesis [9], which leads to a disfunction in adipose tissue and an unhealthy fat mass expansion [20]. In fact, the negative association between body iron (haemoglobin, ferritin and iron) and circulating leptin, which was in agreement with previous studies [21, 22], reflects the negative impact of iron excess in adipose tissue function.

  1. Gao Y, Liu J, Bai Z, et al. Iron down-regulates leptin by suppressing protein O-GlcNAc modification in adipocytes, resulting in decreased levels of O-glycosylated CREB. J Biol Chem 2019;294:5487-5495.
  2. Gao Y, Li Z, Gabrielsen JS, et al. Adipocyte iron regulates leptin and food J Clin Invest 2015;125:3681-3691.

Round 2

Reviewer 2 Report

The authors of the present review article have satisfactorily responded to my comments and suggestions and made the necessary changes to the revised paper. Therefore, there are no further comments.

Author Response

The authors are grateful for the reviewer comments which have contributed to clarify the message of our paper and to improve the quality of our submission.

Reviewer 3 Report

I have to admit that the authors improved the quality of the manuscript. However, markers of AT function like adiponectin or others are still lacking. Leptin is not a functional marker, it is generally secreted proportionally to fat mass, so the correlations are similar to those obtained to fat depots.

Author Response

We agree with the reviewer in that leptin is not strictly a functional marker. However, there is information in the literature how excess iron affects leptin mRNA in in vitro studies. We now translate this information to human plasma. The following changes are now made in the discussion:

“Through different mechanisms, high hepcidin and high serum ferritin levels are a sign of an excess iron accumulation inside adipose tissue cells. Hepcidin functions as a regulator of intestinal iron absorption, iron recycling and mobilization from iron stores [18], while serum ferritin is used as a measurement of iron stores and high values of ferritin are a sign of iron overload [19]. This excess iron has been related to adipocyte hypertrophy and lower adipogenesis [9], which leads to a dysfunction in adipose tissue and an unhealthy fat mass expansion [20]. In the current study, the negative association between markers body iron (haemoglobin, ferritin and iron) and circulating leptin, in agreement with previous studies [21, 22], might reflect the negative impact of iron excess on healthy fat mass expansion. Supporting this idea, circulating adiponectin, an established marker of adipose tissue function, was negatively correlated with circulating ferritin and adipocyte iron excess in humans and mice [6,7]. In addition, treatment of 3T3-L1 adipocytes with iron decreased leptin mRNA in a dose-dependent manner [22]. ”